# CODA-19: Using a Non-Expert Crowd to Annotate Research Aspects on 10,000+ Abstracts in the COVID-19 Open Research Dataset

**Ting-Hao (Kenneth) Huang**[1], **Chieh-Yang Huang**[1] **Chien-Kuang Cornelia Ding**[2],
**Yen-Chia Hsu**[3], **C. Lee Giles**[1]

[1]Pennsylvania State University, University Park, PA, USA
`{txh710,chiehyang,clg20}@psu.edu`
[2]University of California, San Francisco, CA, USA. `Cornelia.Ding@ucsf.edu`
[3]Carnegie Mellon University, Pittsburgh, PA, USA. `yenchiah@andrew.cmu.edu`

## Abstract

This paper introduces **CODA-19**[1], a human-annotated dataset that codes the **Background, Purpose, Method, Finding/Contribution, and Other** sections of 10,966 English abstracts in the COVID-19 Open Research Dataset. CODA-19 was created by 248 crowd workers from Amazon Mechanical Turk within 10 days, achieving a label quality comparable to that of experts. Each abstract was annotated by nine different workers, and the final labels were obtained by majority vote. The inter-annotator agreement (Cohen's kappa) between the crowd and the biomedical expert (0.741) is comparable to inter-expert agreement (0.788). CODA-19's labels have an accuracy of 82.2% when compared to the biomedical expert's labels, while the accuracy between experts was 85.0%. Reliable human annotations help scientists to understand the rapidly accelerating coronavirus literature and also serve as the battery of AI/NLP research, but obtaining expert annotations can be slow. We demonstrated that a non-expert crowd can be rapidly employed at scale to join the fight against COVID-19.

## 1 Introduction

Just as COVID-19 is spreading worldwide, the rapid acceleration in new coronavirus literature makes it hard to keep up with. Researchers have thus teamed up with the White House to release the COVID-19 Open Research Dataset (CORD-19) (Wang et al., 2020), containing 130,000+ related scholarly articles (as of August 13, 2020). The Open Research Dataset Challenge has also been launched on Kaggle to encourage researchers to use cutting-edge techniques to gain new insights from these papers (AI and collaborators, 2020). A

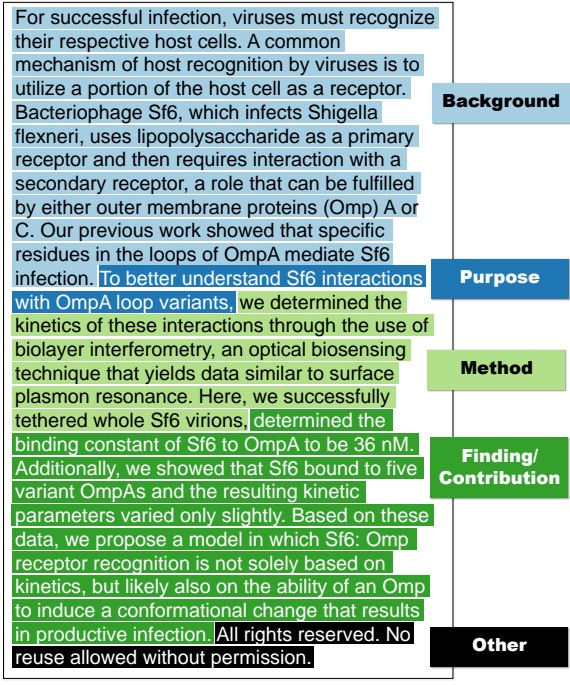

Figure 1: An example of the final crowd annotation for the abstract of (Hubbs et al., 2019).

key knowledge to comprehend scientific papers is that most papers follow a specific *structure* (Alley, 1996), where a set of components are presented in a particular order: a paper typically begins with the background information, such as the motivation and the known facts relevant to the problem, followed by the methods that the authors used to study the problem, and eventually presents the results and discusses the implications (Dasigi et al., 2017). Knowing this structure is essential to comprehend the core arguments and contributions of a paper effectively. Prior work, as shown in Table 1, has proposed various schemes to analyze the structures of scientific articles. Parsing all the CORD-19's papers automatically and represent their structures using a semantic scheme (*e.g.*, background,

---

[1]COVID-19 Research Aspect Dataset (CODA-19): `http://CODA-19.org`

| | Corpus | Document type | Instance type | # of documents | # of sentence | Annotator type | # of classes | Public? |
|---|---|---|---|---|---|---|---|---|
| **This work** | **CORD-19** | **abstract** | **clause** | **10,966** | **103,978** | **crowd** | **5** | **yes** |
| Liakata et al. | ART† | paper | sentence | 225 | 35,040 | expert | 18 | yes |
| Ravenscroft et al. | MCCRA† | paper | sentence | 50 | 8,501 | expert | 18 | yes |
| Teufel and Moens | CONF | paper | sentence | 80 | 12,188 | expert | 7 | no |
| Kim et al. | MEDLINE | abstract | sentence | 1,000 | 10,379 | expert | 6 | no |
| Contractor et al. | PubMed | paper | sentence | 50 | 8,569 | expert | 8 | no |
| Morid et al. | UpToDate+PubMed | mixed | sentence | 158 | 5,896 | expert | 8 | no |
| Banerjee et al. | CONF+arXiv | paper | sentence | 450 | 4,165 | expert | 3 | no |
| Huang and Chen | NTHU database | abstract | sentence | 597 | 3,394 | expert | 5 | no |
| Agarwal and Yu | BioMed Central | paper | sentence | 148 | 2,960 | expert | 5 | no |
| Hara and Matsumoto | MEDLINE | abstract | sentence | 200 | 2,390 | expert | 5 | no |
| Zhao et al. | JOUR | mixed | sentence | ... | 2,000 | expert | 5 | no |
| McKnight and Srinivasan | MEDLINE | abstract | sentence | 204 | 1,532 | ... | 4 | no |
| Chung | PubMed | abstract | sentence | 318 | 829 | expert | 4 | no |
| Wu et al. | CiteSeer | abstract | sentence | 106 | 709 | expert | 5 | no |
| Dasigi et al. | PubMed | section | clause | 75 | <4,497* | expert | 7 | no |
| Ruch et al. | PubMed | abstract | sentence | 100 | ... | ... | 4 | no |
| Lin et al. | PubMed | abstract | sentence | 49 | ... | expert | 4 | no |

Table 1: Comparison of datasets, excluding structured abstracts. Abbreviations CONF and JOUR mean conference and journal papers, respectively. The dagger symbol † means that the corpus is self-curated. The "mixed" document type means that the dataset contains instances from both full papers and abstracts. Symbol "..." means unknown. The last "Public" column means if the dataset is publicly downloadable on the Internet. The symbol * means that the work only provides the number of clauses (sentence fragments).

method, result, etc.) will make it easier for both humans and machines to comprehend and process the information in these 13,000+ papers.

Modern automated language understanding approaches often require large-scale human annotations as training data to reach good performance levels. Researchers traditionally relied on experts to annotate the structures of scientific papers, as shown in Table 1. However, producing such annotations for thousands of papers will be a prolonged process if we only employ experts, whose availability is much more limited than that of non-expert annotators. As a consequence, most of the human-annotated corpora that labeled the structures of scientific articles covered no more than one thousand papers (Table 1). Obtaining expert annotations can be too slow to respond to COVID-19, so we explore an alternative approach: using **non-expert crowds**, such as workers on Amazon Mechanical Turk (MTurk), to produce high-quality, useful annotations for thousands of scientific papers.

Researchers have used non-expert crowds to annotate text, for example, for machine translation (Wijaya et al., 2017; Gao et al., 2015; Yan et al., 2014; Zaidan and Callison-Burch, 2011; Post et al., 2012), natural language inference (Bowman et al., 2015; Khot et al., 2018), or medical report analysis (Maclean and Heer, 2013; Zhai et al., 2013; Good et al., 2014; Li et al., 2016). While domain

experts are still valuable in creating high-quality labels (Stubbs, 2013; Pustejovsky and Stubbs, 2012), and concerns have also been raised over the uses of MTurk (Fort et al., 2011; Cohen et al., 2016), employing non-expert crowds has been shown to be an effective and scalable approach to create datasets. However, annotating *papers* is still often viewed as an "expert task". Majority of the datasets only used experts (Table 1) or the information provided by the paper authors (Table 2) to denote the structures of scientific papers. One exception was the SOLVENT project by Chan *et al.* (2018). They recruited MTurk workers to annotate tokens in paper abstracts with the research aspects (*e.g.*, Background, Mechanism, Finding). However, while the professional editors recruited from Upwork performed well, the MTurk workers' token-level accuracy was only 59%, which was insufficient for training good machine-learning models.

This paper introduces **CODA-19**, the **COVID-19 Research Aspect Dataset**, presenting the first outcome of our exploration in using non-expert crowds for large-scale scholarly article annotation. CODA-19 contains 10,966 abstracts randomly selected from CORD-19. Each abstract was segmented into sentences, which were further divided into one or more shorter text fragments. All 168,286 text fragments in CODA-19 were labeled with a "research aspect," *i.e.*, **Background, Pur-**

| | Corpus | Document | Instance | # of documents | # of sentence | # of classes | Public? |
|---|---|---|---|---|---|---|---|
| Dernoncourt and Lee | PubMed | structured abstract | sentence | 195K | 2.2M | 5 | yes |
| Jin and Szolovits | PubMed | structured abstract | sentence | 24K | 319K | 7 | yes |
| Huang et al. | MEDLINE | structured abstract | sentence | 19K | 526K | 3 | no |
| Boudin et al. | PubMed | structured abstract | sentence | 260K | 349K | 3 | no |
| Banerjee et al. | PubMed | structured abstract | sentence | 20K | 216K | 3 | no |
| Chung | PubMed | structured abstract | sentence | 13K | 156K | 4 | no |
| Shimbo et al. | MEDLINE | structured abstract | sentence | 11K | 114K | 5 | no |
| McKnight and Srinivasan | MEDLINE | structured abstract | sentence | 7K | 90K | 4 | no |
| Chung and Coiera | MEDLINE | structured abstract | sentence | 3K | 45K | 5 | no |
| Hirohata et al. | MEDLINE | structured abstract | sentence | 683K | ... | 4 | no |
| Lin et al. | MEDLINE | structured abstract | sentence | 308K | ... | 4 | no |
| Ruch et al. | PubMed | structured abstract | sentence | 12K | ... | 4 | no |

Table 2: Comparison of datasets, leveraging structured abstracts that do not require human-labeling effort. Some works that involve human-labeled data are also present in Table 1.

**pose, Method, Finding/Contribution, or Other**. This annotation scheme was adapted from SOL-VENT (Chan et al., 2018), with minor changes. Figure 1 shows an example annotated abstract.

In our project, 248 crowd workers from MTurk were recruited and annotated the whole CODA-19 **within ten days**.[2] Each abstract was annotated by nine different workers. We aggregated the crowd labels for each text segment using majority voting.

The resulting crowd labels had a label accuracy of 82% when compared against the expert labels on 129 abstracts. The inter-annotator agreement (Cohen's kappa) was 0.741 between the crowd labels and the expert labels, while it was 0.788 between two experts. We also established several classification baselines, showing the feasibility of automating such annotation tasks.

## 2 Related Work

A significant body of prior work have explored the space of revealing or parsing the structures of scientific articles, including composing structured abstracts (Hartley, 2004), identifying argumentative zones (Teufel et al., 1999; Mizuta et al., 2006; Liakata et al., 2010), analyzing scientific discourse (de Waard and Maat, 2012; Dasigi et al., 2017; Banerjee et al., 2020), supporting paper writing (Wang et al., 2019; Huang and Chen, 2017), and representing papers to reduce information overload (de Waard et al., 2009). In this section, we review the datasets that were created to study the structures of scientific articles.

We categorize all the datasets that denoted the structures of scientific papers into two categories:

*(i)* the datasets that used human labors to manually annotate the sentences in scientific articles (Table 1), and *(ii)* the datasets that leveraged the structured abstracts (Table 2).

In the first category, the researchers recruited a group of annotators— often experts, such as medical doctors, biologists, or computer scientists— to manually label the sentences in papers with their research aspects (*e.g.*, background, method, finding). CODA-19 belongs to the first category. Table 1 reviews the existing datasets in this category. To the best of our knowledge, only two other datasets can be download from the Internet besides our work. Nearly all of the datasets of this kind were annotated by domain experts or researchers, which thus limited their sizes significantly. In Table 1, CODA-19 is the only dataset that contains more than one thousand papers. Our work presents a scalable and efficient solution that employs non-expert crowd workers to annotate scientific papers. Furthermore, our labels were based on clauses (also referred to as sentence fragments or sub-sentences), which provide more detailed information than the majority of other works that used sentence-level annotations.

In the second category, researchers used the section titles that came with structured abstracts in scientific databases (*e.g.*, PubMed) to label sentences. A structured abstract is an abstract with distinct, labeled sections (*e.g.*, Introduction, Methods, Results) (Hartley, 2004). Different journals have different guidelines for section titles. To form a coherent and standardized dataset, researchers often mapped these different titles into a smaller set of labels. The sizes of the datasets in the second category (Table 2) were typically larger because they did not require extra annotating effort. This

---
[2]From April 19, 2020 to April 29, 2020, including the time for worker training and post-task survey.

line of research is inspiring, however, assigning the same label to all the sentences in the same section overlooks the information granularity at the sentence level. Furthermore, not every journal uses the format of structured abstracts. The language used for describing a research work with a coherent paragraph might defer from the language used for presenting the work with a set of predetermined sections. Our work creates an in-domain dataset with high-quality labels for each sentence fragment in the 10,000+ abstracts, regardless of their formats.

## 3 CODA-19 Dataset Construction

CODA-19 has 10,966 abstracts that contain a total of 2,703,174 tokens and 103,978 sentences, which were divided into 168,286 segments. The data is released as a 80/10/10 train/dev/test split.

### 3.1 Annotation Scheme

CODA-19 uses a five-class annotation scheme to denote research aspects in scientific articles: **Background, Purpose, Method, Finding/Contribution, or Other**. Table 3 shows the full annotation guidelines we developed to instruct workers. We updated and expanded this guideline daily during the annotation process to address workers' questions and feedback. This scheme was adapted from SOLVENT (Chan et al., 2018), with three changes. First, we added an "Other" category. Articles in CORD-19 are broad and diverse (Colavizza et al., 2020), so it is unrealistic to govern all cases with only four categories. We are also aware that CORD-19's data came with occasional formatting or segmenting errors. These cases were also to be put into the "Other" category. Second, we replaced the "Mechanism" category with "Method." Chan *et al.* created SOLVENT with the aim of discovering the analogies between research papers at scale. Our goal was to better understand the contribution of each paper, so we decided to use a more general word, "Method," to include the research methods and procedures that cannot be characterized as "Mechanisms." Also, biomedical literature widely used the word "mechanism," which could also be confusing to workers. Third, we modified the name "Finding" to "Finding/Contribution" to allow broader contributions that are not usually viewed as "findings."

### 3.2 Data Preparation

We used Stanford CoreNLP (Manning et al., 2014) to tokenize and segment sentences for all the abstracts in CORD-19. We further used comma (,), semicolon (;), and period (.) to split each sentence into shorter fragments, where a fragment has no fewer than six tokens (including punctuation marks) and has no orphan parentheses.

As of April 15, 2020, 29,306 articles in CORD-19 had a non-empty abstract. An average abstract had 9.73 sentences (SD = 8.44), which were further divided into 15.75 text segments (SD = 13.26). Each abstract had 252.36 tokens (SD = 192.89) on average. We filtered out the 538 (1.84%) abstracts with only one sentence because many of them had formatting errors. We also removed the 145 (0.49%) abstracts that had more than 1,200 tokens to keep the working time for each task under five minutes (see Section 3.4). We randomly selected 11,000 abstracts from the remaining data for annotation. During the annotation process, workers informed us that a few articles were not in English. We identified these automatically using langdetect[3] and excluded them.

### 3.3 Interface Design

Figure 2 shows the worker interface, which we designed to guide workers to read and label all the text segments in an abstract. The interface showed the instruction on the top (Figure 2a) and presented the task in three steps: In Step 1, the worker was instructed to spend ten seconds to take a quick glance at the abstract. The goal was to get a high-level sense of the topic rather than to fully understand the abstract. In Step 2, we showed the main annotation interface (Figure 2b), where the worker can go through each text segment and select the most appropriate category for each segment one by one. In Step 3, the worker can review the labeled text segments (Figure 2c) and go back to Step 2 to fix any problems.

### 3.4 Annotation Procedure

**Worker Training and Recruitment** We first created a qualification Human Intelligence Task (HIT) to recruit workers on MTurk ($1/HIT). The workers needed to watch a five-minute video to learn the scheme, go through an interactive tutorial to learn the interface, and sign a consent form to

---

[3]langdetect: https://github.com/Mimino666/langdetect

| Aspect | Annotation Guideline |
|---|---|
| **Background** | "Background" text segments answer one or more of these questions:
• Why is this problem important?
• What relevant works have been created before?
• What is still missing in the previous works?
• What are the high-level research questions?
• How might this help other research or researchers? |
| **Purpose** | "Purpose" text segments answer one or more of these questions:
• What specific things do the researchers want to do?
• What specific knowledge do the researchers want to gain?
• What specific hypothesis do the researchers want to test? |
| **Method** | "Method" text segments answer one or more of these questions:
• How did the researchers do the work or find what they sought?
• What are the procedures and steps of the research? |
| **Finding/ Contribution** | "Finding/Contribution" text segments answer one or more of these questions:
• What did the researchers find out?
• Did the proposed methods work?
• Did the thing behave as the researchers expected? |
| **Other** | • Text segments that do *not* fit into any of the four categories above.
• Text segments that are *not* part of the article.
• Text segments that are *not* in English.
• Text segments that contain *only* reference marks (*e.g.*, "[1,2,3,4,5") or dates (*e.g.*, "April 20, 2008").
• Captions for figures and tables (*e.g.* "Figure 1: Experimental Result of ...")
• Formatting errors.
• Text segments the annotator does not know or is not sure about. |

Table 3: CODA-19's annotation guideline for crowd workers.

obtain the qualification. We granted custom qualifications to 400 workers who accomplished the qualification HIT. Only the workers with this qualification could do our tasks.[4]

**Posting Tasks in Smaller Batches**    We divided 11,000 abstracts into smaller batches, where each batch has no more than 1,000 abstracts. Each abstract forms a single HIT. We recruited nine different workers through nine assignments to label each abstract. Our strategy was to post one batch at a time. When a batch was finished, we assessed its data quality, sent feedback to workers to guide them, or blocked workers who constantly had low accuracy before proceeding with the next batch.

**Worker Wage and Total Cost**    We aimed to pay an hourly wage of $10. The working time of an abstract was estimated by the average reading speed of English native speakers, *i.e.*, 200-300 words per minute (Siegenthaler et al., 2012). For an abstract, we rounded up ($\#token/250$) to an integer as the estimated working time in minutes and paid ($0.05 +$ Estimated Working Minutes $\times$ $0.17) for it. As a result, 59.49% of our HITs were priced at $0.22, 36.41% were at $0.39, 2.74% were at $0.56,

0.81% were at $0.73, and 0.55% were at $0.90. We posted nine assignments per HIT. Adding the 20% MTurk fee, coding each abstract (using nine workers) cost $3.21 on average.

In this project, each worker received an average of ($3.21/9)/1.2 = $0.297 for annotating one abstract. We empirically learned that the CS Expert (see Section 4) spent an average of 50.8 seconds (SD=10.4, N=10) to annotate an abstract, yielding an estimated hourly wage of $0.297 \times (60 \times 60/50.8) = $21.05$; and the MTurk workers in SOLVENT took a median of 1.3 minutes to annotate one abstract (Chan et al., 2018), yielding an estimated hourly wage of $0.297 \times (60/1.3) = $13.71$. We thus believe that the actual hour wage for workers were close to or over $10.

## 3.5    Label Aggregation

The final labels in CODA-19 were obtained by majority voting over crowd labels, excluding the labels from blocked workers. For each batch of HITs, we manually examined the labels from workers who frequently disagreed with the majority-voted labels (Section 3.4). If a worker had abnormally low accuracy or was apparently spamming, we retracted the worker's qualification to prevent him/her from taking future tasks. We excluded the labels from these removed workers when aggregating the final

---

[4]Four built-in MTurk qualifications were also used: Locale (US Only), HIT Approval Rate ($\geq$98%), Number of Approved HITs ($\geq$3000), and the Adult Content Qualification.

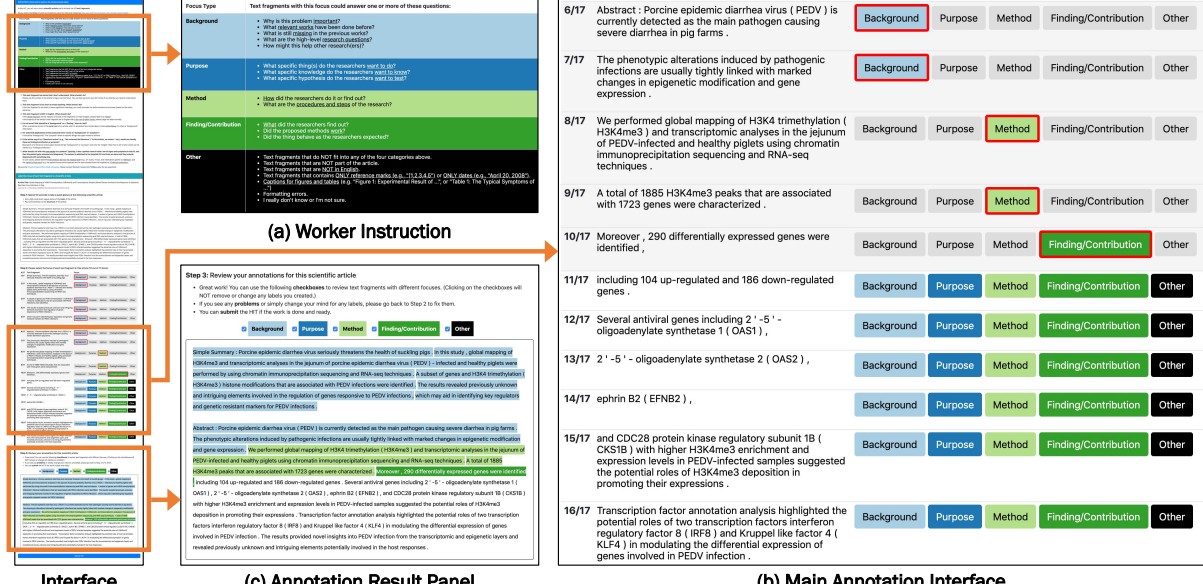

Figure 2: The worker interface used to construct CODA-19.

labels. Note that there can be ties when two or more aspects received the same highest number of votes (*e.g.,* 4/4/1 or 3/3/3). We resolved ties by using the following tiebreakers, in order: Finding, Method, Purpose, Background, Other.

## 4 Data Quality Assessment

We worked with a biomedical expert and a computer scientist to assess label quality; both experts are co-authors of this paper. The biomedical expert (the "Bio" Expert in Table 4) is an MD and also a PhD in Genetics and Genomics. She is now a resident physician in pathology at the University of California, San Francisco. The other expert (the "CS" Expert in Table 4) has a PhD in Computer Science and is currently a Project Scientist at Carnegie Mellon University.

Both experts annotated the same 129 abstracts randomly selected from CODA-19. The experts used the same interface as that of the workers (Figure 2). We used scikit-learn's implementation (Pedregosa et al., 2011) to compute the inter-annotator agreement (Cohen's kappa). The kappa between the two experts was 0.788. Table 4 shows the aggregated crowd label's accuracy, along with the precision, recall, and F1-score of each class. CODA-19's labels have an accuracy of 0.82 and a kappa of 0.74 when compared against the two experts' labels. It is noteworthy that when we compared labels between the two experts, the accuracy (0.850) and kappa (0.788) were only slightly higher. The crowd workers performed best in labeling "Background" and "Finding," and they had nearly perfect precision for the "Other" category. Figure 3 shows the normalized confusion matrix for the aggregated crowd labels versus the biomedical expert's labels. Many "Purpose" segments were mislabeled as "Background," which might indicate more ambiguous cases between these two categories. During the annotation period, we received several emails from workers asking about the distinctions between these two aspects. For example, does "potential applications of the proposed work" count as "Background" or "Purpose"?

## 5 Classification Baselines

We further examined machines' capacity for annotating research aspects automatically. Seven baseline models were implemented: Linear SVM, Random Forest, Multinomial Naive Bayes (MNB), CNN, LSTM, BERT, and SciBERT.

**Data Preprocessing** The tf-idf feature was used for Linear SVM and Random Forest. We turned all words into lowercase and removed those with frequency lower than 5. The final tf-idf feature contained 16,775 dimensions. Two variations of feature were used for MNB, the n-gram counts feature and the n-gram tf-idf feature. Using grid search method, the n-gram counts feature combining unigram, bigram, and trigram with minimum frequency of 3 yielded the best result. The final n-gram feature contained 181,391 dimensions. The

| Eval. Label | Gold Label | Background | | | Purpose | | | Method | | | Finding | | | Other | | | acc | kappa |
|---|---|---|---|---|---|---|---|---|---|---|---|---|---|---|---|---|---|---|
| | | P | R | F1 | P | R | F1 | P | R | F1 | P | R | F1 | P | R | F1 | | |
| **Crowd** | Bio | .827 | .911 | .867 | .427 | .662 | .519 | .783 | .710 | .744 | .874 | .838 | .856 | .986 | .609 | .753 | .822 | .741 |
| **Crowd** | CS | .846 | .883 | .864 | .700 | .611 | .653 | .818 | .633 | .714 | .800 | .931 | .860 | .986 | .619 | .761 | .821 | .745 |
| **CS** | Bio | .915 | .966 | .940 | .421 | .746 | .538 | .670 | .785 | .723 | .958 | .789 | .865 | .867 | .852 | .860 | .850 | .788 |

Table 4: Crowd performance using both Bio Expert and CS Expert as the gold standard. CODA-19's labels have an accuracy of 0.82 and a kappa of 0.74, when compared against two experts' labels. It is noteworthy that when we compared labels between two experts, the accuracy (0.850) and kappa (0.788) were only slightly higher.

| Model | Background | | | Purpose | | | Method | | | Finding | | | Other | | | Accuracy |
|---|---|---|---|---|---|---|---|---|---|---|---|---|---|---|---|---|
| | P | R | F1 | P | R | F1 | P | R | F1 | P | R | F1 | P | R | F1 | |
| **# Sample** | 5062 | | | 821 | | | 2140 | | | 6890 | | | 562 | | | 15475 |
| **SVM** | .658 | .703 | .680 | .621 | .446 | .519 | .615 | .495 | .549 | .697 | .729 | .712 | .729 | .699 | .714 | .672 |
| **RF** | .671 | .632 | .651 | .696 | .365 | .479 | **.716** | .350 | .471 | .630 | **.787** | .699 | .674 | .742 | .706 | .652 |
| **MNB-count** | .654 | .714 | .683 | .549 | .514 | .531 | .570 | .585 | .577 | .711 | .691 | .701 | **.824** | .425 | .561 | .665 |
| **MNB-tfidf** | .655 | .683 | .669 | .673 | 391 | .495 | .640 | .469 | .541 | .661 | .754 | .704 | .757 | .383 | .508 | .659 |
| **CNN** | .649 | .706 | .676 | .612 | .512 | .557 | .596 | .562 | .579 | .726 | .702 | .714 | .743 | .795 | .768 | .677 |
| **LSTM** | .655 | .706 | .680 | **.700** | .464 | .558 | .634 | .508 | .564 | .700 | .724 | .711 | .682 | .770 | .723 | .676 |
| **BERT** | .719 | .759 | .738 | .585 | **.639** | .611 | .680 | .612 | .644 | .777 | .752 | .764 | .773 | **.874** | .820 | .733 |
| **SciBERT** | **.733** | **.768** | **.750** | .616 | .636 | **.626** | .715 | **.636** | **.673** | **.783** | .775 | **.779** | .794 | .852 | **.822** | **.749** |

Table 5: Baseline performance of automatic labeling using the crowd labels of CODA-19. SciBERT achieves highest accuracy of 0.749 and outperforms other models in every aspects.

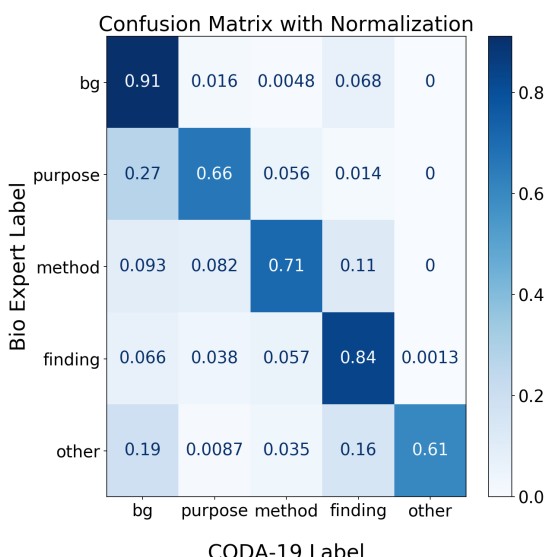

Figure 3: The normalized confusion matrix for the CODA-19 labels versus the biomedical expert's labels.

**Models** Machine-learning approaches were implemented using Scikit-learn (Pedregosa et al., 2011) and deep-learning approaches were implemented using PyTorch (Paszke et al., 2019). The following are the training setups.

- **Linear SVM:** We did a grid search for hyper-parameters and found that $C = 1$, $tol = 0.001$, and *hinge loss* yielded the best results.

- **Random Forest:** With the grid search, 150 estimators yielded the best result.

- **Multinomial Naive Bayes (MNB):** Several important early works used Naive Bayes models for text classification (Rennie et al., 2003; McCallum et al., 1998). When using n-gram counts as the feature, the default parameter, $alpha = 1.0$, yielded the best result. For the one using n-gram tf-idf feature, $alpha = 0.5$ yielded the best result.

- **CNN:** The classic CNN (Kim, 2014) was implemented. Three kernel sizes (3, 4, 5) were used, each with 100 filters. The word embedding size was 256. A dropout rate of 0.3 and L2 regularization with weight $1e - 6$ were used when training. We used the Adam optimizer, with a learning rate of $5e - 5$. The model was trained for 50 epochs and the one

n-gram tf-idf feature combining unigram, bigram, and trigram with minimum frequency of 10 yielded the best result where the dimensions is 41,966. For deep-learning approaches, the vocabulary size was 16,135, where tokens with frequency lower than 5 were replaced by <UNK>. Sequences were padded with <PAD> if containing less than 60 tokens and were truncated if containing more than 60 tokens.

with highest validation score was kept for testing.

- **LSTM:** We used 10 LSTM layers to encode the sequence. The encoded vector was then passed through a dense layer for classification. The word embedding size and the LSTM hidden size were both 256. The rest of the hyperparameter and training setting was the same as that of the CNN model.

- **BERT:** Hugging Face's implementation (Wolf et al., 2019) of the Pretrained BERT (Devlin et al., 2018) was used for fine-tuning. We fine-tuned the pretrained model with a learning rate of $3e - 7$ for 50 epochs. Early stopping was used when no improvement occurred in the validation accuracy for five consecutive epochs. The model with the highest validation score was kept for testing.

- **SciBERT:** Hugging Face's implementation (Wolf et al., 2019) of the Pretrained SciBERT (Beltagy et al., 2019) was used for fine-tuning. The fine-tuning setting is the same as that of the BERT model.

**Result** Table 5 shows the results for the six baseline models: SciBERT preformed the best in overall accuracy. When looking at each aspect, all the models performed better in classifying "Background," "Finding," and "Other," while identifying "Purpose" and "Method" was more challenging.

## 6 Discussion

Annotating scientific papers was often viewed as an "expert task" that is difficult or impossible for non-expert annotators to do. Many datasets that labeled scientific papers were thus produced by small groups of experts. For example, two researchers manually created the ACL RD-TEC 2.0, a dataset that contains 300 scientific abstracts (QasemiZadeh and Schumann, 2016); a group of annotators "with rich experience in biomedical content curation" created MedMentions, a corpus containing 4,000 abstracts (Mohan and Li, 2019); and many datasets used in biomedical NLP shared tasks were manually created by the organizers and/or their students, such as the ScienceIE in SemEval'17 (Augenstein et al., 2017) and Relation Extraction in SemEval'18 (Gábor et al., 2018). Our work sug-

gests that non-expert crowds can be used for these types of data-labeling tasks.

Prior work often used crowd workers to annotate pieces of lower-level information on papers or medical documents, such as images (Heim et al., 2018) or named entities (*e.g.,* medical terms (Mohan and Li, 2019), disease (Good et al., 2014), medicine (Abaho et al., 2019).) Our work shows that crowd workers, to certain extent, can comprehend the high-level structures and discourses in papers, and therefore could be assigned with more complex, higher-level tasks.

## 7 Conclusion and Future Work

This paper introduces CODA-19, a human-annotated dataset that codes the Background, Purpose, Method, Finding/Contribution, and Other sections of 10,966 English abstracts in the COVID-19 Open Research Dataset. CODA-19 was created by a group of MTurk workers, achieving a label quality comparable to that of experts. We demonstrated that a non-expert crowd can be rapidly employed at scale to join the fight against COVID-19.

One future direction is to improve classification performance. We evaluated the automatic labels against the biomedical expert's labels, and the SciBERT model achieved an accuracy of 0.774 and a Cohen's kappa of 0.667, indicating some space for further improvement. Furthermore, one motivation for spotting research aspects automatically is to help search and information extraction (Teufel et al., 1999). We have teamed up with the group who created COVIDSeer[5] to explore the possible uses of CODA-19 in such systems.

## Acknowledgments

This project is supported by the Huck Institutes of the Life Sciences' Coronavirus Research Seed Fund (CRSF) at Penn State University and the College of IST COVID-19 Seed Fund at Penn State University. We thank the crowd workers for participating in this project and providing useful feedback. We thank VoiceBunny Inc. for granting a 60% discount for the voiceover for the worker tutorial video to support projects relevant to COVID-19. We also thank Tiffany Knearem, Shih-Hong (Alan) Huang, Joseph Chee Chang, and Frank Ritter for the great discussion and useful feedback.

---

[5]CovidSeer: https://covidseer.ist.psu.edu/

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
