# OpenReview forum: "CODA-19: Using a Non-Expert Crowd to Annotate Research Aspects on 10,000+ Abstracts in the COVID-19 Open Research Dataset"
_aclweb.org/ACL/2020/Workshop/NLP-COVID — NLP-COVID-2020_

### Official Review · AnonReviewer1 · 2020-06-03
**Potentially important topic, but a number of questionable claims and an overall lack of contact with relevant literature detract from the paper quite a bit.**

**Rating:** 4
**Confidence:** 4

**Review:**

OVERALL

Thank you for the opportunity to review this interesting paper about annotation of abstracts from the CORD-19 corpus.

OVERALL STRENGTHS:

1.The paper is written in the official language of the conference.

2. The paper is within the page limits of the conference.

3. The abstract accurately reflects the contents of the paper.

4. The discussion of worker wages and total cost is very nice. The paper could do a better job of showing the reader why it is important by putting it into the context of Fort et al.’s paper on the ethics of using AMT for NLP data (see the References section below), as well as Cohen et al.’s paper on the relationship between Turker pay rates and actual Turker earnings (from that work, there appears not to be one).  Additionally, the paper mentions a target hourly wage, but does not report the ACTUAL hourly wage.  This seems like an omission.

5. The data quality assessment is very nice.

OVERALL AREAS FOR IMPROVEMENT:

For visibility, consider adding “COVID-19” to the title of the paper.

The literature review is missing quite a bit of relevant work. This is an important gap in the paper because adequate coverage of related work would help in making the significance and innovation arguments.

There are a couple of claims in the Introduction that are either not relevant or not supportable, and coming in the Introduction, they reduce a reader’s confidence in the paper as a whole. Specifics:

The statement that producing annotations for thousands of papers can be a prolonged process if only expert annotators are used and that their availability is limited is not connected with the rest of the paper.  There is also no evidence that it is true.  In fact due to the virus and attendant shutdown, expert annotators might have actually been MORE available at the time this work was done than at any time since the beginning of corpus annotation. I don’t have data on that, but the paper doesn’t present any data for the opposing hypothesis, either, and since there probably is none available, I would advise simply dropping any critique of manual annotation by experts.  If the authors feel that it must be included, then be balanced--point out the benefits of expert annotation (see, for example, Amber Stubbs’s dissertation on the topic--sorry I couldn’t find the reference for this one; I suggest contacting her directly).

The claim that data sparsity is a major challenge for biomedical text mining is not supported, and probably not supportable.  There is actually a LOT of data available for BioNLP research--that is why most people got into the field in the first place (see Cohen and Demner-Fushman’s book “Biomedical natural language processing)”. Tons of projects have produced quite a bit of data--e.g., Cohen et al.’s paper on multi-model annotation in the CRAFT corpus lists 25 (twenty-five) biomedical corpus constructions just in the five years before the publishing of that paper.

The scientific premises of the paper include the idea that the annotation task would typically be thought of as requiring expert knowledge; that is why the paper uses it to study the use of non-expert annotators.  The approach to such a research question is reasonable, but the data and the task are not suitable to answering the question.  The annotation scheme (Background/Purpose/etc.) does not seem like one that is necessarily difficult for non-experts, for example in the case where they are marked explicitly in abstracts. So, either demonstrate that they’re not marked, or improve the argumentation with a reasoned claim that the task would EVER be thought of as requiring expert annotators.

The paper’s description of the dataset is inadequate.  It samples the CORD-19 corpus, but the CORD-19 corpus has a number of versions (commercial, non-commercial, for example), and the paper does not tell us from which part the CODA corpus was sampled.

In Table 2, it is not clear to me that F1 is the right measure of agreement here.  Consider supporting an argument that it is.

The paper reports kappa, but does not say how P(e)n was calculated.  That makes it difficult to interpret the kappa score at all.

The result given at the end of Section 5 regarding which model gave the “best” classification performance is not relevant.  The point of reporting classification scores in a paper on a corpus is as a measure of annotation quality and quantity--the argument would be that if a classifier can build a good model, then the annotations are probably consistent and the quantity of data is probably adequate (see the Pustejovsky and Stubbs reference).  Which model performs “the best” is not actually well-evaluated in the paper, and it is not relevant to the paper, so I would strongly suggest removing that small paragraph altogether.

As academic courtesy if nothing else, I strongly recommend that you mention the classic work on crowdsourcing for production of NLP data.

REFERENCES (sorry for all of the Cohen references--he has done a lot of work in this area)

Fort, Karën, Gilles Adda, and K. Bretonnel Cohen. "Amazon mechanical turk: Gold mine or coal mine?." Computational Linguistics 37.2 (2011): 413-420.

Cohen, K. Bretonnel, et al. "Ethical issues in corpus linguistics and annotation: Pay per hit does not affect effective hourly rate for linguistic resource development on amazon mechanical turk." LREC...

International Conference on Language Resources & Evaluation:[proceedings]. International Conference on Language Resources and Evaluation. Vol. 2016. No. W40. NIH Public Access, 2016.

Cohen, Kevin Bretonnel, and Dina Demner-Fushman. Biomedical natural language processing. Vol. 11. John Benjamins Publishing Company, 2014.

Cohen, K. Bretonnel, Karin Verspoor, Karën Fort, Christopher Funk, Michael Bada, Martha Palmer, and Lawrence E. Hunter. "The Colorado Richly Annotated Full Text (CRAFT) corpus: Multi-model annotation in the biomedical domain." In Handbook of Linguistic Annotation, pp. 1379-1394. Springer, Dordrecht, 2017.

Pustejovsky, James, and Amber Stubbs. Natural Language Annotation for Machine Learning: A guide to corpus-building for applications. " O'Reilly Media, Inc.", 2012.

---

> ### Author Response · Authors · 2020-08-17
> **Revision Submitted**
>
> We just submitted a new version of the paper, in which we addressed most of your comments. Please let us know if you have more questions. Thanks!

---

### Official Review · AnonReviewer2 · 2020-06-09
**Fairly sound approach, but several serious concerns**

**Rating:** 5
**Confidence:** 4

**Review:**

This manuscript discusses the use of crowdsourcing to label sub-sentence segments with labels that look a lot like the ones used for structured abstracts.

With few exceptions, I found the manuscript to be clear. The task is a reasonably interesting one (though it would be a good idea to motivate it a bit more by showing what it could be used for) and the methods are reasonable and I am reasonably convinced that the approach described would produce the results shown. This paper is a pretty good example of the sort of work that can be done with crowdsourcing. I am not aware of other work approaching similar classification tasks with crowdsourcing, but I did not go looking. One very important point that seems to be buried is the fact that the annotation took 10 days. Every annotation project I have ever been involved with that used experts took months; this is a strong argument in favor of this approach.

There are quite a few things that concern me, however.

First, the methods are all straightforward applications of fairly well known work. For the machine learning that is ok, but there are few insights in the crowdsourcing methods or the results that would not be strongly expected. In other words the authors need to make much more clear any claims to originality or significance.

The paper neglects to mention that many of the abstracts in CORD-19 are structured. For example, the abstract for document e9cgnhbq (PMID 20153051) has the following sections: Objective, Methods, Results, Conclusion. While the correlation may not be 1-to-1 to the annotation scheme, it seems that the mapping might be pretty high. How much does this help the crowd workers? Would it help the classifiers?

The claim that data sparsity is a challenge in general is of course true - there will never be enough annotated data to train a system to do every task we could want at near-human performance levels - but somewhat irrelevant. If there are no existing datasets that can be used to perform this task, then it is a challenge and the comments about the difficulty of annotating for other tasks is irrelevant. If there are existing datasets for the task needed, then the challenge is not lack of data but whether the dataset really does allow the task to be addressed. More likely, there are datasets that solve a similar task, and these should be evaluated and tested.

Which version of CORD-19 was used? A new version is released every week.

While it is appropriate to provide compensation that roughly approximates 'fair', it is not mentioned how long each abstract took on average and therefore what the actual pay rate was. It would be useful to hear how long it took the experts as well. When I do annotation - admittedly for a more complex task - I almost always have to read the text multiple times.

This is essentially a text classification task, a task that multinomial Naive Bayes performs quite well on (probably not as well as BERT, but still). See:
McCallum, Andrew, and Kamal Nigam. "A comparison of event models for naive bayes text classification." AAAI-98 workshop on learning for text categorization. Vol. 752. No. 1. 1998.
Tackling the poor assumptions of naive bayes text classifiers." Proceedings of the 20th international conference on machine learning (ICML-03). 2003.).

There is a strong but imperfect ordering of the sections: "Background" will almost always come before "Method". Note that at least one of the cited papers mentions this: Huang 2017. So why was that information not used in the classification models? The way to use it would be different for most of the models, but for naive bayes, the way to use it would be to switch to using a hidden markov model, which handles sequential information.

Structured abstracts provide a great deal of information that is not being exploited here. In addition to being used as a feature directly, they could be used to train a second classifier that would serve as a feature: the structured abstracts will provide section labels and your existing TF-IDF vectorizer provides samples of vectors from segments. If these are run over a large corpus - not even necessarily covid-related - then a multinomial naive bayes model (or your favorite high dimensional instance classifier) could be created that predicts the section label from the TFIDF vector (importantly, this would work even if the abstract being predicted was not structured). There would be some section labels that are rare, these can simply be removed from the model. Then any TF-IDF vector for a segment could have section labels predicted, with that prediction fed as a feature for the classification of your annotation scheme.

There is no analysis provided to show whether 9x annotation is better than, say 5x annotation for this task. There is no analysis provided of how large the corpus needs to be, ie an ablation study with the classifiers. If the classifiers have the same performance on 11k abstracts that they would have on 5.5k abstracts, then maybe we don't need 11k. If 5x annotation of 5.5k abstracts provides close enough performance, then a sufficient corpus could have been created for 1/4 the cost.

It is not made clear why the annotation scheme will help fight COVID-19: what use cases do research aspects enable?

---

> ### Author Response · Authors · 2020-08-17
> **Revision Submitted**
>
> We just submitted a new version of the paper, in which we addressed most of your comments. Please let us know if you have more questions. Thanks!

---

### Public Comment · ~Berry_de_Bruijn2 · 2020-05-20
**Very readable article: well written, appropriately designed, and the resources are of significant use to the community.**

Very readable article: well written, appropriately designed, and the resources are of significant use to the community.
Somewhat ironically, it is not entirely clear where the Methods section ends, the Results section starts, and where the Discussion sits. Due to the two-fold purpose (that is, evaluating the crowd-sourced annotation first, and then the ML classifiers), additional methods are introduced after results of the earlier effort are discussed. Section captions are not entirely helpful, and it would be better to stick with a standard label set for those: there is no clear Discussion section, nor a Conclusions section. The manuscript would be helped if it were reworked within a tighter structure.

---

### Comment · Program_Chairs · 2020-06-23
**Revise and resubmit**

Based on feedback from the reviewers and public comments, we would invite you to Revise the manuscript in response to the reviewers' feedback. (NB: at this stage this does not need to be finalized ahead of the workshop, but soon after.)

The objectives of the work are related to describing the details of a new resource, and assessing its quality. This is important information for anyone who might want to work with the data set. However, there are some concerns about the presentation in the manuscript, and some details related to the task definition and the contextualization of the work can improve the contribution.

We would also like you to present this work at the workshop on July 09 and will be in touch about that.

Thank you for your submission!

---

> ### Author Response · Authors · 2020-06-23
> **Thank you**
>
> Thanks. We're happy to present this paper at the workshop!

---

### Decision · Program_Chairs · 2020-10-15

**Decision:**

Accept

**Comment:**

Formally recording the decision previously communicated.